# Cognitive Behavioral Therapy for Migraine Headache: A Systematic Review and Meta-Analysis

**DOI:** 10.3390/medicina58010044

**Published:** 2021-12-28

**Authors:** Ji-yong Bae, Hyun-Kyung Sung, Na-Yoen Kwon, Ho-Yeon Go, Tae-jeong Kim, Seon-Mi Shin, Sangkwan Lee

**Affiliations:** 1Department of Cardiology and Neurology of College of Korean Medicine, Kyung Hee University, Seoul 02447, Korea; giraffe124@naver.com; 2Department of Korean Pediatrics, College of Korean Medicine, Semyung University, Jecheon 27136, Korea; shksolar@gmail.com (H.-K.S.); mintypink@semyung.ac.kr (T.-j.K.); 3Department of Obstetrics and Gynecology, College of Korean Medicine, Ga-Chon University, Seongnam-si 13120, Korea; darnabal@naver.com; 4Department of Oriental Internal Medicine, College of Oriental Medicine, Semyung University, Jecheon 27136, Korea; kohoyeon@gmail.com; 5Department of Internal Korean Medicine, College of Korean Medicine, Semyung University, Jecheon 27136, Korea; bunggujy21@hanmail.net; 6Clinical Trial Center, Wonkwang University Gwangju Hospital, Gwangju 61729, Korea; 7Department of Oriental Internal Medicine and Neuroscience, Oriental Medical School, Wonkwang University, Iksan 54538, Korea

**Keywords:** migraine, migraine headache, headache, cognitive behavioral therapy, systematic review

## Abstract

*Background and Objectives*: Migraine headaches are chronic neurological diseases that reduce the quality of life by causing severe headaches and autonomic nervous system dysfunction, such as facial flushing, nasal stuffiness, and sweating. Their major treatment methods include medication and cognitive behavioral therapy (CBT). CBT has been used for pain treatment and various psychogenic neurological diseases by reducing pain, disability, and emotional disorders caused by symptoms of mental illness and improving the understanding of mental health. This study aimed to evaluate the effectiveness and safety of CBT in treating migraines. *Materials and Methods*: Seven electronic databases were searched from the date of inception to December 2020. Randomized controlled studies (RCTs) using CBT as an intervention for migraine were included. The primary outcome of this study was to determine the frequency of migraines and the intensity of migraines on Visual Analog Scale (VAS), the frequency of drug use, Migraine Disability Assessment (MIDAS), and Headache Impact Test (HIT-6) index. The two authors independently conducted the data extraction and quality assessment of the included RCTs, and conducted meta-analysis with RevMan V.5.4. *Results*: Among the 373 studies, 11 RCTs were included in this systematic review. Seven out of the 11 RCTs were conducted in the USA, and four were conducted in the UK, Germany, Iran, and Italy, respectively. Headache frequency and MIDAS scores were statistically significant reduced. In the subgroup analysis, headache strength was significantly reduced. Two of the included studies reported adverse effects, including worsening of migraine intensity and frequency, respiratory symptoms, and vivid memory of a traumatic event. *Conclusions*: CBT for migraine effectively reduced headache frequency and MIDAS score in meta-analysis and headache intensity subgroup analysis, with few adverse events. Additional RCTs with CBT for migraine headaches are needed for a more accurate analysis.

## 1. Introduction

Migraine is a disease characterized by severe headache accompanied by symptoms, such as nausea, photophobia, phonophobia, and vomiting [1]. The prevalence of migraine is estimated to be 15–18% [2], and it is two to three times higher in women than in men [3,4]. Migraine is a chronic neurological disorder and autonomic nervous system dysfunction that affects patients’ quality of life [5,6]. The preferred treatment for migraine is medication administration. Acute medications include paracetamol, non-steroidal anti-inflammatory drugs (NSAIDs), and triptans [7]. Routine use of opioids and barbiturates is not preferred because of their poor safety and tolerability [8,9,10].

Overuse of painkillers can lead to a variety of side effects and medication overuse headache; therefore, attention should be paid to drug abuse during migraine symptoms [11,12]. Non-drug therapy is known to have fewer side effects and can be used simultaneously with medications [12,13].

Physicians who treat migraines are increasingly interested in complementary treatments [14,15]. Medications, such as antidepressants, hypertension treatments, and flunarizine, were common treatment for migraines. However, acupuncture and biobehavial therapy are also used to prevent migraine headaches for patients with little response to existing drug treatment or pregnant women, or patients with psychological disorders. [7,16]. Nutritional supplements, such as riboflavin, pyridoxine, folate, cobalamin, and vitamin D, have recently been widely used as preventive treatments for migraine [17,18,19].

Cognitive behavioral therapy (CBT) is a treatment that uses cognitive factors to improve mental disorders and psychological distress [20]. The latest practice guidelines emphasize CBT as a selective psychotherapy for problems ranging from depression, anxiety, and personality disorders to chronic pain, addiction, and relationship pain [21]. Previous studies have demonstrated the effectiveness of behavioral therapy for migraine headaches, including CBT, relaxation, and biological feedback, to reduce the frequency of migraine attacks and migraine-related disorders [22,23,24].

Although CBT has been used as a treatment for migraine headaches, there is only one systematic review on pediatric migraine [25], and no systematic review has been conducted on all ages. In this study, the author stated that there is evidence that CBT is viable in the treatment of childhood migraine, and therefore should be provided as a first-line treatment, not only as an add-on if medications are not effective. This study aimed to summarize the results of randomized controlled trials (RCTs) to evaluate the clinical efficacy and safety of CBT on migraine patients of all age. Through this study, we aimed to analyze the availability and effectiveness of CBT in migraine treatment and to help it be used in clinical situations.

## 2. Materials and Methods

### 2.1. Search Strategy and Selection Criteria

Two independent authors (J.-y.B. and N.-Y.K.) searched for randomized controlled trials (RCTs) to evaluate the efficacy of CBT for migraine until October 2020. MEDLINE-PubMed, Embase, and Cochrane Central Register of Controlled Trials (English databases); China National Knowledge Infrastructure (http://www.cnki.net, accessed on 15 October 2021) (Chinese database); and Research Information Service System (http://www.riss.kr, accessed on 15 October 2021), Oriental Medicine Advanced Searching Integrated System (http://oasis.kiom.re.kr, accessed on 15 October 2021), Korean Studies Information Service System (http://kiss.kstudy.com, accessed on 15 October 2021), National Digital Science Library (http://www.ndsl.kr, accessed on 15 October 2021), and DBPIA (http://www.dbpia.co.kr, accessed on 15 October 2021) (Korean databases) databases were searched. “Migraine” and “Cognitive behavioral therapy” were used as search terms, and searching conditions were modified to each database. The details of the search strategies are provided in Appendix A.

This review was registered on PROSPERO (registration number CRD42020223201) and the systematic review was written in accordance with the Preferred Reporting Items for Systematic reviews and Meta-Analyses (PRISMA) 2020 statement [26].

### 2.2. Eligibility Criteria

Only RCTs and quasi-RCTs were included in this review. Eligible participants both had episodic and chronic migraines, regardless of age, sex, and presence of aura. Studies in which the International Classification of Headache Disorders (ICHD) criteria (I, II, and III) and Headache International Society (IHS) criteria were used as diagnostic criteria for migraine were included in this review. The primary outcome was frequency of migraine (per month), intensity of migraine indicated on the visual analogue scale, frequency of taking medication (per month), MIDAS index, and HIT-6 index.

### 2.3. Outcome Assessment and Data Extraction

Two authors (N.-Y.K. and H.-K.S.) assessed the included studies. A detailed analysis was performed by two authors (J.-y.B. and N.-Y.K.) using an extraction form that included country, study design, age, sex, number of participants, inclusion criteria, exclusion criteria, duration, outcome index, effect size, and adverse effects (Table 1). Outcome measures were assessed at baseline, after treatment, and follow-up. Primary outcome measures were headache days assessed by headache diary (mostly defined as a day containing 2 or more hours of headache); headache duration and pain intensity; Migraine Disability Assessment (MIDAS) [27], assessing migraine-related disability, with a higher score reflecting more severe disability; and Headache Impact Test (HIT-6) [28], assessing the impact of headache, score range is 36–78, with a higher score reflecting higher impact and number of days using rescue medication. Various methods of CBT were used in the included studies, and each method is listed in Table 2. If there were any missing or unclear results, we contacted the authors of the included studies.

### 2.4. Assessment of Risk of Bias

The Cochrane risk of bias (ROB) assessment tool was used for quality assessment of the included study [38]. Each bias was classified into three categories: high risk, low risk, and unclear risk and was evaluated by two authors (HKS and TJK).

Seven domains were assessed: sequence generation, allocation concealment, blinding of participants and research personnel, blinding of outcome assessment, incomplete outcome data, selective outcome reporting, and other biases.

### 2.5. Data Synthesis

As a measure of migraine improvement, the weighted mean difference (MD) and standard deviation (SD) of the primary outcomes were calculated for meta-analysis using Review Manager (RevMan) ([Computer program]. Version 5.4. The Cochrane Collaboration, 2020) using 95% confidence intervals (CI), respectively. We assessed effect estimates with MD and standard deviation SD for continuous outcomes. Heterogeneity was assessed by I2 statistics. If I2 was >75%, the random effect model was adopted for meta-analysis; otherwise, the fixed effect model was used.

### 2.6. Heterogeneity and Subgroup Analysis

Subgroup analysis was divided according to the control group’s method. Heterogeneity was assessed using the I^2^ statistic. When heterogeneity was higher than 70%, we also performed subgroup analysis to explain the source of heterogeneity using Review Manager.

## 3. Results

### 3.1. The Results of Literature Search and Screening

A total of 373 studies were identified from the electronic databases, and among these, 233 studies were screened after removing 140 duplicate studies. In the screening, 55 articles without full text were removed, and 178 articles were assessed for eligibility. Excluding 47 articles that were not RCTs and 122 articles that were non-CBT, 11 articles were finally included (Figure 1).

### 3.2. Description of the Included Studies

Seven RCTs [12,24,29,32,34,36,37] were conducted in the USA, and four RCTs [30,31,33,35] were conducted in the UK, Italy, Germany, and Iran, respectively. All included studies were written in English. The total number of patients with migraine analyzed in the review was 621 (intervention: 317, control: 304). A summary of the included studies is presented in Table 1.

There were two types of control groups: one control group received conventional therapy called no treatment/treatment as usual (TAU), and the other group was the sham control group, who underwent lifestyle modification and received placebo behavioral therapy. According to the Western criteria, four trials [30,33,36,37] utilized the IHS criteria but did not mention the version, five [12,29,32,34,35] used the ICHD-II criteria, and two [24,31] used the ICHD-III criteria. CBT methods as interventions were different in detail and are shown in Table 2 and Table 3.

### 3.3. Risk of Bias in the Included Studies

As shown in Figure 2, seven studies reported randomization methods, and there was no statement of randomization method in four studies [30,31,34,37]. Five studies [24,29,32,33,35] used computerized randomization, one study [36] used a random number table, and one study [12] conducted randomization based on the inpatient day of participants. There were three single-blind studies [32,33,36], two double-blind studies [12,29], one without blinding [24], and five without mentioning blinding [30,31,34,35,37]. There were no incomplete data in the three studies [29,31,34], and there were missing data in eight studies. Three studies [30,36,37] showed insufficient information on dropout data, four studies [12,24,32,33] used intention-to-treat analysis (ITT) for missing data, and one study [35] used analysis-per-protocol and not ITT analysis for missing data. There were insufficient data to judge selective reporting without two studies. One study [24] with the original protocol and one study [30] that was thought to have selectively reported on an outcome of their study.

### 3.4. Primary Outcome

#### 3.4.1. Headache Frequency

Six RCTs [24,29,30,31,33,35] reported the mean and standard deviation of the headache frequency, excluding five studies (three studies [32,36,37] that did not mention the mean and standard deviation, one study [12] that reported the median value only, and one study [34] that reported the percentage of migraine days only, instead of mean and SD). In meta-analysis, heterogeneity was high (χ^2^ = 28.18, *p* = 0.0003, I^2^ = 82%) so we used subgroup analysis according to the type of control group. The days of headache per month decreased significantly in sub-group analysis with the education group (*p* < 0.0001) (Figure 3). However, there was no statistically significant difference in headache frequency in the sub-group analysis with the WL/TAU/SMC group (*p* = 0.36) and heterogeneity remained high (χ^2^ = 16.11, *p* = 0.001, I^2^ = 81%) (Figure 4).

#### 3.4.2. Migraine Disability Assessment Score

Five studies [12,24,29,33,34] reported the MIDAS score, excluding six studies that did not mention it. Sub-group analysis was conducted because of clinical heterogeneity. In sub-group analysis compared with the education group, Ped MIDAS score decreased significantly (*p* = 0.02) and showed moderate heterogeneity and showed moderate heterogeneity and showed moderate heterogeneity (χ^2^ = 2.02, *p* = 0.16, I2 = 51%) (Figure 5). In sub-group analysis compared with the WL/TAU/SMC group, MIDAS score also showed a statistically significant decrease (*p* = 0.005) with low heterogeneity (χ^2^ = 0.27, *p* = 0.60, I^2^ = 0%) (Figure 6).

#### 3.4.3. Headache Impact Test Score

Three out of 11 studies [12,32,33] reported the HIT-6 score. In subgroup analysis compared with the WL/TAU/SMC group, HIT-6 score showed significant change (*p* = 0.02) and moderate heterogeneity was observed (χ^2^ = 2.11, *p* = 0.15, I^2^ = 53%) (Figure 7).

### 3.5. Adverse Events

Two studies [24,29] reported adverse effects. Powers reported 199 adverse events (90 in the CBT plus amitriptyline group vs. 109 in the headache education plus amitriptyline group), including status migrainosus or worsening of migraine, respiratory adverse events (e.g., influenza, pneumonia), and other expected adverse effects of amitriptyline (fatigue, drowsiness, and dizziness). Seng reported two adverse events in the MBCT-M group: vivid recollection of a traumatic event while practicing mindfulness and severe increase in headache frequency and pain intensity.

## 4. Discussion

Migraine headache is a chronic neurological disease that varies in its frequency and severity, and [29] is a prevalent condition that can severely affect personal, social, and work life during attacks [30,31]. Although the standard treatment for migraine headaches is currently taking medication, a psychiatric approach with a high level of psychological co-prosperity has also recently drawn attention [28,31]. Individuals with migraine are increasingly approaching complementary and integrative health strategies [14,15]. Because patients have an increased preference for CBT treatment for a variety of reasons, several behavioral treatments for migraine prevention have been used, especially during pregnancy or when pharmacological choices for patients are limited, such as low efficacy or lack of durability in pharmacotherapy, or in combination with pharmacological treatments [16].

CBT refers to cognitive processes related to the development and maintenance of psychopathology, particularly emotional pain and dysfunction, which are primarily conducted during sessions, requiring therapists to coordinate interventions to best help patients [34]. CBT therapy enables patients to develop preventive and acute care strategies, such as trigger identification, modification of maladaptive interrelated thoughts, feelings and behaviors surrounding headache, and physiological autoregulation strategies. Behavioral therapies for migraine headaches, including cognitive behavioral therapy, relaxation, and biofeedback, have demonstrated efficacy in reducing migraine attack frequency and migraine-related disorders [22,23,35,36].

Several previous studies have shown that CBT reduces disability and chronic pain in patients [39,40]. Our study showed that CBT had a significant effect on reducing headache frequency and MIDAS scores, which is consistent with the results of previous studies. Therefore, although many studies with more samples are needed in the future, it is believed that CBT can be considered for its use as a complementary therapy for migraine treatment.

In subgroup analysis of headache frequency, CBT was effective in reducing migraine incidence date compared to education alone, but there was no significant difference in headache frequency compared to the WL/TAU/SMC group. In the analysis of the fixed effect model, the *p* value was 0.004, but the heterogeneity was high at 81%, so it was analyzed using a random effect model for a more conservative analysis.

MIDAS score in subgroup analysis showed significant change compared to both the education (*p* = 0.02) and WL/TAU/SMC group (*p* = 0.005). Additionally, in subgroup analysis with WL/TAU/SMC, HIT-6 score showed significant change (*p* = 0.02). Although the number of studies included in this analysis is small, CBT has been found to have a significant effect on alleviating disability in migraine patients and has shown that it can also be an option as a treatment to reduce disability caused by migraine headaches.

Meta-analysis and subgroup analysis about headache intensity and the number of days using rescue medication were impossible due to the high heterogeneity caused by differences in the control group, and due to the lack of the number of studies

The strength of our study is that it is the first systematic review using meta-analysis of CBT for migraine. Meta-analysis is difficult to use when heterogeneity is significantly high or insufficient data are available. If heterogeneity was high, we conducted a meta-analysis and evaluated it by performing a subgroup analysis to evaluate the efficacy of CBT for migraine. Second, our study screened the results of not only an English-based database but also various database searches, such as Korean and Chinese databases, to reduce publication bias. Finally, for studies with only the F value, we contacted the author to obtain the mean (SD) value. Five out of the two authors replied to our request and sent us the raw data [12,24].

This study has some limitations. Clinical heterogeneity was hypothesized to be high because there was a difference between patients and interventions in each study. There are many types of migraine, such as acute/chronic, episodic, and migraine with or without aura. The study was conducted comprising patients with migraine of various types and intensities. Due to the diversity of participants in each study, there was heterogeneity in the study demographics. In most cases, the sex ratio of the participants was tilted toward women without two studies (one study [30] did not mention it, and one study [37] was male dominant). There were also studies [34,37] that only included children and adolescents. The reason for this participant configuration is that migraine is most prevalent during the most potentially productive and childbearing periods [29,30]. CBT treatment methods were relatively different from study to study, and the limitations of CBT treatment methods were that they should be implemented in part by experts skilled in each method [12,24]. However, some methods have been implemented through methods, such as CD-ROM [34]; thus, we can expect effective individual treatment to be performed through various video media and internet lectures. Most studies [32,33,34,35,37] combined conventional therapies, such as ibuprofen, acetaminophen, NSAIDs, triptans, and muscle relaxants. Participants in some studies [12,29,30,31,32,34] received pharmacological prophylaxis for migraine (topiramate, nortriptyline, propranolol, or amytriptiline according to the physician’s choice based on the patient profile, such as previous failures and contraindications). Hence, it is difficult to conclude whether the symptoms improved only by the effect of CBT. Finally, there were no RCTs on CBT for migraine, which were searched from the Korean and Chinese databases. We hypothesized that taking Western medicine, acupuncture, and herbal medicine has already been standardized in China and Korea to treat migraine; thus, no RCTs using CBT have been conducted. Since CBT is receiving attention as an alternative to migraine treatment, various RCTs should be conducted to evaluate the effectiveness of CBT on migraines combined with existing standard treatments.

## 5. Conclusions

Our study found that CBT can improve headache frequency and MIDAS scores in patients with migraine, with few adverse events. In subgroup analysis, headache intensity in the CBT group had a statistically significant effect on migraine. Further RCTs with CBT for migraine headaches are needed for a more accurate analysis.

## Figures and Tables

**Figure 1 medicina-58-00044-f001:**
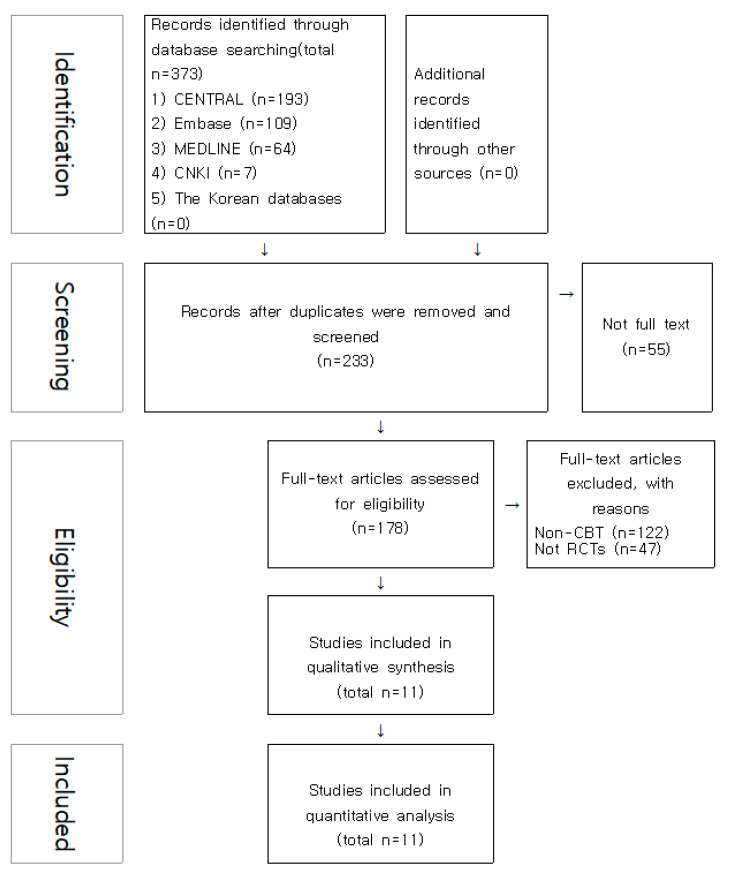
Scheme of the data selection process.

**Figure 2 medicina-58-00044-f002:**
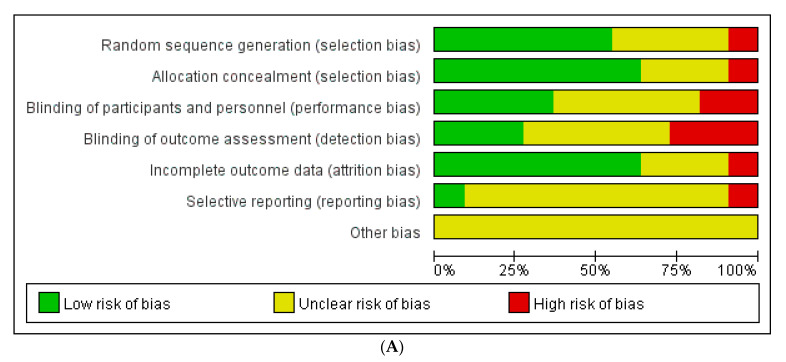
Assessment of risk of bias. (**A**): Risk of bias graph. (**B**): Risk of bias summary.

**Figure 3 medicina-58-00044-f003:**
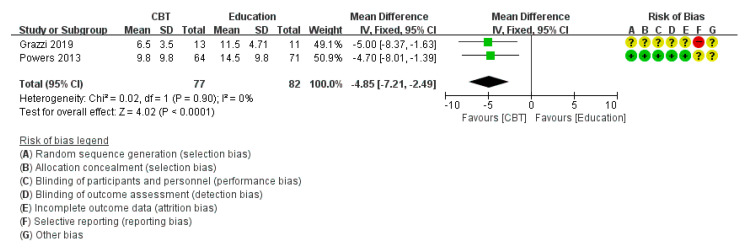
Subgroup analysis of cognitive behavioral treatment compared with the education group regarding the difference from baseline in headache frequency.

**Figure 4 medicina-58-00044-f004:**
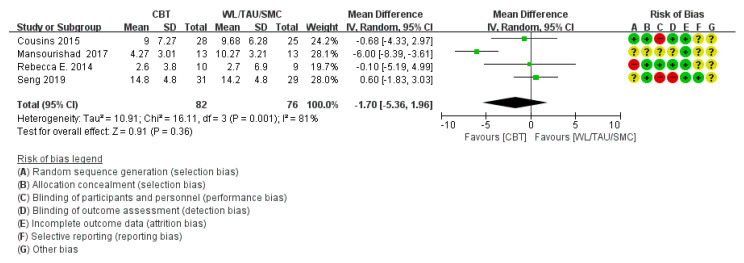
Subgroup analysis of cognitive behavioral treatment compared with the WL/TAU/SMC group regarding the difference from baseline in headache frequency.

**Figure 5 medicina-58-00044-f005:**
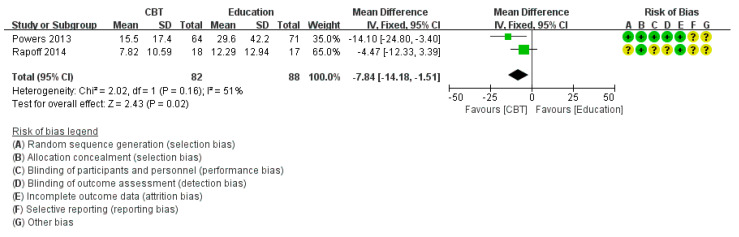
Subgroup analysis of cognitive behavioral treatment compared with the education group regarding the difference from baseline in Pediatric Migraine Disability Assessment score.

**Figure 6 medicina-58-00044-f006:**
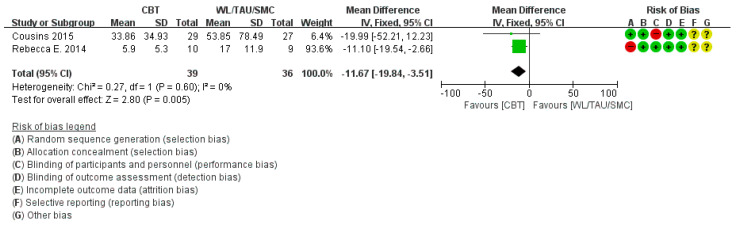
Subgroup analysis of cognitive behavioral treatment compared with the WL/TAU/SMC group regarding the difference from baseline in Pediatric Migraine Disability Assessment score.

**Figure 7 medicina-58-00044-f007:**
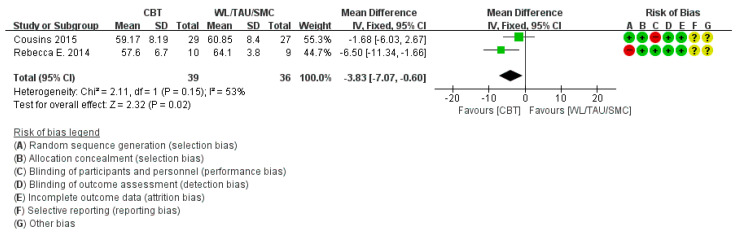
Subgroup analysis of cognitive behavioral treatment compared with the WL/TAU/SMC group for difference from baseline in Headache Impact Test (HIT-6) score.

**Table 1 medicina-58-00044-t001:** Basic characteristics of the included studies.

Study ID (Year, Country)	Study Design	Age (Years, Mean ± SD)	Sex (n, Male/Female)	Inclusion Criteria	Exclusion Criteria	Intervention (n)	Control (n)	Duration	Follow-Up
Powers [29] (2020, USA)	RCT	I: 14.4 ± 1.9 C: 14.4 ± 2.1	I: 13/51 C: 15/56	(1) ICHD-II - ≩15 days HA per month (HA diary) (2) PedMIDAS >20 - at least moderate disability	(1) Medication Overuse (ICHD-II) (2) Current use of amitriptyline or other prophylactic antimigraine medication within a period equivalent to less than 5 half-lives before study screening, (3) other chronic pain condition -fibromyalgia -complex regional pain syndrome II (4) Abnormal electrocardiogram (5) Severe orthostatic intolerance or dysregulation, (6) Documented developmental delay or impairment (7) Severe psychiatric comorbidity (eg, psychosis, bipolar disorder, major depressive disorder) (8) PedMIDAS > 140 (excessive disability and need for multisystemic therapies) (9) pregnancy or being sexually active without use of medically accepted form of contraception (barrier or hormonal methods) (10) use of disallowed medications including opioids, antipsychotics, antimanics, barbiturates, benzodiazepines, muscle relaxants, sedatives, tramadol, or herbal products.	CBT + Amitriptyline (64)	Education + Amitriptyline (71)	20 weeks	3, 6, 9 months
Seng [24] (2019, USA)	RCT	I: 36.2 ± 10.6 C: 44.2 ± 11.5	I: 2/29 C: 3/26	(1) ICHD-3 (2) Headache days ≥ 6 per month (3) Aged 18–65 years (4) Ability to read English (5) capacity to consent	(1) Continuous headache over the course of 30 days (2) Initiation of preventive migraine treatment within 4 weeks from baseline or during study (3) Severe psychiatric illness - active suicidality - active psychosis - falling cognitive screen (4) Inability to adhere to headache diary (recorded < 26/30 days)	MBCT-M (31)	WL/TAU (29)	8–10 weeks (8 sessions)	0, 1, 2, 4 months
Grazzi [30] (2019, Italy&USA)	RCT	I: 42.1 ± 11.6 C: 41.8 ± 11,1	Not mentioned	(1) Age 18–65 years (2) High-frequency migraine w/o aura according to the IHS-beta-2013 criteria (3) No withdrawal intervention during 18 months	None	ACT + Education + Prophylaxis (13)	Education + Prophylaxis (11)	6 weeks	3, 6, 12 months
Mansourishad [31] (2017, Iran)	RCT	I: 33.6 ± 6.2 C: 30.7 ± 5.2	I: 0/13 C: 0/13	(1) ICHD-III criteria w or w/o aura (2) Age 20–40 years (3) Female (4) Minimum of 6 month gap between dignosis and beginnng of the study (5) Minimum of moderate IQ (6) No other psychological therapies over past 8 months (7) No use of medications for anxiety or depression during the past 3 months	(1) Severe physical illness (2) Serious neurological disorders or symptoms of psychosis (3) Unwillingness to continue treatment (4) Risk of suicidal thoughts and attempts requiring urgent intervention	MBCT (13)	Control (13)	6 weeks	
Smitherman [32] (2016, USA)	RCT	I: 29.6 ± 13.4 C: 32.1 ± 12.8	I: 1/15 C: 2/13	Chronic Migraine comorbid insomnia (1) ICHD-II w/o MOH (2) ICSD-3 for insomnia	(1) Secondary headache dosorder including MOH (2) Pregnancy or breastfeeding (3) Being unable to read or speak English at a 6th grade level (4) Untreated sleep apnea (5) Active alcohol or substance use or dependence (6) Active bipolar disorder (7) Psychiatric hospitalization within last year (8) Employment involving rotating shift work schedule (9) Recent or expected change in preventive headache pharmacotherapy	CBTi (16)	Sham control (Lifestyle modification) (15)	Baseline 2 weeks + 6 weeks (biweekly)	2, 6 weeks after Tx
Cousins [33] (2015, UK)	RCT	I: 40.67 ± 12.79 C: 37.97 ± 12.04	I: 8/28 C: 5/32	(1) 18–75 years (2) Onset >6 months ago (3) Day >3 HA days per month - assessed by HA diary - including episodic, chronic HA (4) IHS 2nd criteria - w or w/o aura	(1) Secondary HA (physical conditions lilely to cause HA) (2) Pregnancy (3) Current psychotic illness (4) Substance dependency (not including headache rescue medication) (5) Currently undergoing psychological therapy (6) inability to complete self-report measures	CBT + Relaxation + SMC (37)	SMC (36)	5 weeks	2, 4 months
Rapoff [34] (2014, USA)	RCT	I: 10.2 ± 2.0 C: 10.2 ± 1.5	I: 8/10 C: 2/15	(1) ICHD-II (w or w/o aura) (2) 7–12 years (3) Migraine occurring on the average at least once per week (by parental or child report and separated by symptom-free periods)	(1) Secondary headaches (2) Mental health condition or was receiving concurrent psychotherapy (3) CBC were in the clinical range at baseline (4) Average HA frequency <1 per week(over a 14-day period)	CD-Rom Headstrong (18)	CD-Rom Education (17)	4 weeks	3 months
Wells [12] (2014, USA)	RCT	I: 45.9 ± 17 C: 45.2 ± 12	I: 1/9 C: 1/8	(1) Migraine w or w/o aura (ICHD-II criteria) (2) 4–14 migraine days/month (3) ≥1 yr history of migraines (4) ≥ 18 years old (5) No additional disease	(1) Current regular meditation/yoga (2) Major systemic illness or unstable medical psychiatric condition (eg, suicide risk) (3) MOH (ICHD-II) (4) Current/planned pregnancy or breastfeeding (5) New prophylactic migraine medicine started within 4 weeks of the screening visit (6) Unwilling to maintain stable migraine medication dosages (7) Failure to complete baseline headache logs	MBSR (10)	TAU (9)	8 weeks	
Fritche [35] (2010,Germany)	RCT	I: 47.7 ± 8.9 C: 48.4 ± 10.1	I: 5/74 C: 9/62	(1) ICHD-II w/o MOH (2) Migraine w and w/o aura (3) Combined HA (migraine + tension-type HA) - if migraine was main headache (4) 18–65 years (5) Intake - triptans on >4 and <10 days per month or - analgesics on >7 and <14 days per month during the past 3 months -combined triptans + analgesics ≨15 intake days including a maximum of 9 triptan intake days	(1) Significant psychiatric disorder (2) Additional secondary headache (3) Additional chronic pain diseases with pharmacological treatment (4) Insufficient knowledge of the German language. (5) Pregnancy	MCT + TAU (79)	Bibliotherapy (information brochures) + TAU (71)	5 weeks	3, 12–30 months
Calhoun [36] (2007, USA)	RCT	I: 33.5 C: 35.0	I: 0/23 C: 0/20	(1) IHS criteria (2) Adult women	(1) Non-pregnant (2) Non-lactating (3) Diagnosis of a primary sleep disorder	BSM (23)	Sham control (Placebo behavioral group) (20)	2 + 6 weeks	6, 12 weeks
Scarff [37] (2002, USA)	RCT	I (HWB): 13.3 ± 2.5 C-1 (HCB): 13.2 ± 2.0 C-2 (WLC): 12.0 ± 2.7	I (HWB): 9/4 C-1 (HCB): 5/6 C-2 (WLC): 10/2	(1) 7–17 years (2) IHS criteria w or w/o aura (3) No Primary medical condition and a negative neurological exam (4) Nor taking daily preventative medication for headaches (5) Average migraine ≥1 per week or ≥ 5 days per month	None	HWB (13)	(1) HCB (11) (2) WLC (12)	6 weeks	3, 6 months

I, intervention; C, control; CBT, cognitive behavioral therapy; SMC, standard medical care; D-CBT-I, Digital-Cognitive Behavioral Therapy for; ISI, Insomnia Severity Index; SDIH-R, Structured Diagnostic Interview and Headache checklist; IHS, International Headache Society; HWB, hand-warming biofeedback; HCB, hand-cooling biofeedback; WLC, wait-list control; MBCT-M, mindfulness cognitive behavioral therapy; WL/TAU, wait-list/treatment as usual; ACT, acceptance and commitment therapy; TAU, treatment as usual; MCT, minimal contact training.

**Table 2 medicina-58-00044-t002:** CBT methods of the included studies.

Study ID	Treatment	Methods
Powers [29] (2020, USA)	I: CBT	Based on coping skills for pediatric pain, modified to include a biofeedback component. (thermal and electromyographic monitoring of the relaxation response)
	C: Education	Consisted of discussion of headache-related education topics
Seng [24] (2019, USA)	MBCT	MBCT protocols by Day et al. [4,5,6,7,8,9,10,11,12,13,14,15,16,17,18,19,31] (1) Automatic pilot: Practice (Body scan, Mindful eating), Homework (Body scan) (2) Awareness of appraisals and stress: Practice (Body scan, Awareness of appraisals, Awareness of thoughts arising during breathing meditation), Homework (Body scan, Mindfulness breath, Awareness of thoughts, Stress events calendar) (3) Mindfulness of the breath: Practice (Breathing space, Labeling automatic thoughts), Homework (Sitting meditation and body scan, Breathing space) (4) Recognizing aversion: Practice (Mindful movement; walking and stretching), Homework (Sitting meditation and mindful movement, Breathing space) (5) Allowing/Letting be: Practice (Identifying automatic thoughts, Sitting meditation with acceptance), Homework (Sitting meditation, Breathing space) (6) Thoughts are not facts: Practice (Sitting meditation, Mindful observation of cognitions and considering alternatives, Awareness of pleasant events), Homework (Choose your own meditation, Breathing space for coping, Pleasant events calendar (7) How can I best take care of myself?: Practice (Sitting meditation, Linking activity and mood/stress/migraine, Identifying warning signs for stress and migraine, Making a plan for nourishing activities), Homework (Develop routine to practice mindfulness, Dealing with stress and migraine) (8) Using Mindfulness to cope with migraine: Practice (Body scan, Relapse prevention, Focused meditation)
Grazzi [30] (2019, Italy&USA)	ACT	○ Objective: patients were trained to practice mindfulness and pain management - psychological flexibility by cultivating six different positive capacities: acceptance, diffusion, sense of self, mindfulness, values, committed actions. - These capacities can improve mental and physical states, disability, and impact in pain conditions (resilience, avoidance, behavior, acceptance). ○ Sessions involved psycho education, discussions, experiential exercises, and home assignments. (1) Session 1: Creative helplessness; the problem of control (2) Session 2: Identifying values; introduction to mindfulness (3) Session 3: Actions guided by values; working with thoughts (4) Session 4: Working with acceptance/willingness (5) Session 5: Committed action; self-as-contest (6) Session 6: Integration; working with obstacles, wrap-up. (7) Final sessions 7/8: Integration; working with what was learned, exercises, wrap-up, practice at home.
	Education	Education of patients, followed by pharmacological prophylaxis for migraine - topiramate, or propanolol, or amytriptiline according to the physician’s choice based on the patient profile, such as previous failures and contraindications.
Mansourishad [31] (2017, Iran)	MBSR	- The sessions focused on developing nonjudgmental thinking and present-moment awareness of thoughts, emotions, and environment. - Homework assignments with the aid of a guided audio file included daily mindfulness meditation practices, such as body scan and breath awareness. - Session-by-session description of the protocol is available in the MBCT manual
Smitherman [32] (2016, USA)	CBTi	(1) Session 1: Included a detailed overview and rationale of the treatment components with instructions for daily home practice. (2) Sessions 2, 3 - Entailed reviewing daily diaries and treatment adherence since the last session, reinforcing progress, and problem-solving around any obstacles to adherence. - Participants continued daily self-monitoring throughout treatment and were instructed to continue practicing their 5 treatment instructions after treatment concluded - For insomnia (1) Go to bed only when sleepy and intending to sleep. (2) Leave the bedroom if unable to sleep after 20 min and return only when sleepy. (3) Use the bedroom only for sleep and sexual activity. (4) Set an alarm and rise daily at the same time. (5) Restrict your time in the bed to your total sleep time plus 30 min. - Sleep hygiene: promoting healthy sleep behaviors and sleep-conducive environmental conditions (e.g., limiting caffeine and meals prior to bedtime, keeping a comfortable bedroom temperature).
	Sham control	Lifestyle modification (1) Eat dinner at a consistent time every evening. (2) Do acupressure (as instructed) for at least 2 min twice daily, once on awakening and once before going to bed. (3) Record all liquids consumed for 3 consecutive “typical” days (identity of liquid, quantity, and time of day) and thereafter keep a consistent liquid intake each day. (4) Do 5 min of stretching/range of motion exercises upon awakening. (5) Consume at least one serving of protein within one hour of arising in the morning (e.g., egg, cheese, cottage cheese, and tofu).
Cousins [33] (2015, UK)	I: CBT +Relaxation C: SMC	(1) Week 1: [Session 1] Introducing the concept of links between thoughts, feelings, symptoms, and behaviors, thought monitoring and relaxation techniques. - Expectations of treatment, Brief assessment, What is CBT, Introduction of headache/thought diary and relaxation techniques (2) Week 2: Telephone follow up: Reviews progress, offer support (3) Week 3: [Session 2] Problem solving and cognitive restructuring including alternative thinking - Review of headache/thought diaries and progress with relaxation techniques - Introduction of problem solving and cognitive restructuring (4) Week 4: Telephone follow up: Reviews progress, offer support (5) Week 5: [Session 3] building on alternative thinking techniques and covered relapse prevention - Review the progress with relaxation/cognitive restructuring techniques (using headache/thought diary) - Problem solving - Ways to maintain improvement
Wells [12] (2014, USA)	MBSR	MBSR protocol by Dr. Jon Kabat-Zinn [7,8,9,10,11,12,13,14,15] - begins with mindfulness of breathing, mindful eating, and the body scan. - subsequent classes build on these practices and slowly add in the other meditative practices. - The all-day retreat includes elements of all the mindfulness practices. - The instructor also gives information about stress and stress relief during the fourth class.
Rapoff [34] (2014, USA)	I: Headstrong	(1) Week 1: Headache education & cognitive behavioral model of pain - Introduction, Types of headache, Prevalence of headache, Features of headache, How headache is diagnosed, The pain puzzle, Headache triggers (2) Week 2: Relaxation - Rationale for relaxation, How to use guided imagery, How to use deep breathing, How to use progressive muscle relaxation (3) Week 3: Cognitive restructuring - Rationale for coping, Thought-out changing, Problem-solving (4) Week 4: Pain behaviors - Positive and negative pain behaviors, Importance of keeping active, Review of all lesson
	C: Education	(1) Week 1: Headache education (Introduction, Types of headache, Prevalence of headache, Features of headache, How headache is diagnosed) (2) Week 2: Cognitive-behavioral model of pain (Introduction to the pain puzzle, puzzle piece-1; nociception, 2; thoughts) (3) Week 3: Cognitive-behavioral model of pain (puzzle piece-3; feelings, 4; behavior) (4) Week 4: Headache triggers (Introduction to headache triggers, key headache triggers, diet, and sleep)
Fritche [35] (2010, Germany)	MCT	(1) Unit 1: Introduction and syndrome education - information about symptoms, pathophysiology and pathopsychology of migraine as well as instructions for progressive muscle relaxation (PMR) (2) Unit 2: Medication rules and the risk of Medication Overuse Headache - information about acute and prophylactic migraine medication and MOH-symptoms and pathomechanisms - establish a clear behavioral intake algorithm in migraine attack situations, (3) Unit 3: Medication intake behavior - aimed at raising awareness for ‘external’ (e.g., availability of drugs, stock-keeping, iatrogenic risk factors like doctor shopping) and ‘internal’ (e.g., fear of attack and losing social functioning, stress level in private and professional life) influences on patient’s medication intake behavior. (4) Unit 4: General and personal risk factors for drug intake - established a general risk profile of medication overuse for each patient. (5) Unit 5: Everyday transfer - aim to establish individual goals for future drug intake and learning how to make use of social support to control intake behavior. (6) Daily PMR and Headache Diary: Daily exercise of PMR as well as keeping a daily headache diary had to be performed during the time between all five sessions (7) Homework: after each session, patients were given topic-related homework.
Calhoun [36] (2007, USA)	BSM Instructions	(1) Schedule consistent bedtime that allows 8 h in bed (2) Eliminate TV, reading, music in bed (3) Use visualization technique to shorten time to sleep onset (4) Move supper ≥4 h before bedtime; limit fluids within 2 h of bedtime (5) Discontinue naps
	Sham Instructions	(1) Schedule consistent suppertime that varies <1 h from day to day (2) Perform acupressure as instructed for 2 min twice daily (3) Record liquid consumption for 3 consecutive days (4) Do 5 min of gentle range of motion exercises every morning (5) Have 1 protein serving at breakfast
Scarff [37] (2002, USA)	I: CBT + HWB C-1: CBT + HCB C-2: WLC	(1) Cognitive behavioral stress management training - younger children (under 13 years) were taught thought stopping and positive self-statements. - Older children were trained to identify stressful thoughts that triggered migraine episodes, test the logic of their thoughts in a more formal manner. (2) Thermal biofeedback training(hand-warming) (3) Progressive muscle relaxation (4) Imagery training of warm places and vasodilation (5) instruction in deep breathing techniques

SMC, standard medical care; MBCT, mindfulness-based cognitive behavioral therapy; ACT, acceptance and commitment therapy; TAU, treatment as usual; IAPT, improving access to psychological therapies; HWB, hand-warming biofeedback; HCB, hand-cooling biofeedback; CBTi, cognitive behavioral therapy for insomnia; BSM, behavioral sleep modification; MBSR, mindfulness-based stress reduction; ACT, acceptance and commitment therapy; MCT, minimal contact training.

**Table 3 medicina-58-00044-t003:** Summary of the results and adverse effects of the included studies.

Study ID	Main Result		Baseline (BL)	Post-Treatment (PT)	Follow-Up (F/U)		*p*-Value				Adverse Effect	Conclusion
			Intervention (Mean ± SD)									
			Control (Mean ± SD)									
Powers [29] (2013)							Group MD in change score at PT				I:90 C:109	○ In I group, headache days and PedMIDAS decreased significantly compared with control group. ○ Headache days were ≥50% Reduction in 66% of I and 36% of C at PT (odds ratio, 3.5 [95%CI, 1.7 to 7.2]; *p* < 0.001). ○ PedMIDAS <20 points were in 75% of I, and 56% of C at PT (odds ratio, 2.4 [95%CI, 1.1 to 5.1]; *p* = 0.02).
	HA frequency (day/month)	I	21.3 ± 5.2	9.8 ± 9.8	Not mentioned		*p* = 0.002					
		C	21.3 ± 5.2	14.5 ± 9.8								
	PedMIDAS	I	68.2 ± 31.7	15.5 ± 17.4	Not mentioned		14.1 (95% CI 3.3, 24.9) *p* = 0.01					
		C	68.2 ± 31.7	29.6 ± 42.2								
Seng [24] (2019)					4 M						○ No AE in control group ○ 2 AE in intervention group (1) Vivid recollection of traumatic event while practicing mindfulness (2) Severe increase in Headache frequency and pain intensity	○ HDI change: the group*month interaction was significant, *p* = 0.004 ○ MIDAS: Group*month (N.S. accounting for divided alpha) *p* = 0.027 ○ MIDI: −0.6/10(I), +0.3/10(C) *p* = 0.007 ○ Headache days, headache intensity: Group*month interaction and group*time interaction (N.S.)
	HDI ^1^	I	52.5 ± 21.2		38.2 ± 16.6		*p* < 0.004					
		C	50.2 ± 16.2		50.4 ± 14.3							
	MIDAS ^2^	Group*month interaction BL vs. 4 M		B = 1.6, 95%CI = −0.7,3.9 F(3,213) = 3.12, *p* = 0.027								
	MIDAS-A ^3^			B = 6.3, 95%CI = −2.0,14.5, F(3,94.6) = 1.21, *p* = 0.312								
	MIDAS-B ^4^			B = 0.3, 95%CI = −0.4,1.1, F(3,102.9) = 0.65, *p* = 0.589								
	HA days/month	I	16.5 ± 6.0		14.8 ± 4.8		*p* = 0.773					
		C	15.5 ± 5.9		14.2 ± 4.8							
	Average Attack Intensity /month ^5^	I	1.7 ± 0.3		1.6 ± 0.3		*p* = 0.888					
		C	1.8 ± 0.3		1.7 ± 0.3							
	Average MIDI/month ^6^	I	2.8 ± 1.6		1.7 ± 2.7		*p* = 0.007					
		C	3.4 ± 2.0		4.4 ± 1.3							
Grazzi [30] (2019)					3 M						N	○ Headache days and medication intake days declined in I group, not in C group
	HA days/month	I	10 ± 2.0		6.5 ± 3.5		N					
		C	9.27 ± 3.43		11.5 ± 4.71							
	Using medication days/month	I	9.4 ± 2.75		5.75 ± 3.3		N					
		C	9.9 ± 3.6		10.5 ± 5.8							
Mansourishad [31] (2017)					3 M						N	○ Covariance analysis showed I group is effective compared with C group in reducing headache frequency (*p*= 0.001< 0.05), duration (*p* = 0.001 < 0.05), and severity (*p*= 0.001 < 0.05) in women with migraine.
	HA Frequency (days/month)	I	10.63 ± 6.16	4.27 ± 3.01	4.73 ± 2.01		*p* = 0.001					
		C	10.81 ± 4.56	10.27 ± 3.21	10.45 ± 6.07							
	HA Intensity	I	6.20 ± 2.30	4.12 ± 1.90	4.32 ± 1.13		*p* = 0.001					
		C	6.41 ± 3.40	6.40 ± 2.83	6.50 ± 2.75							
	HA Duration (h/month)	I	10.63 ± 3.05	5.90 ± 4.29	5.08 ± 2.76		*p* = 0.001					
		C	11.73 ± 5.49	12.45 ± 6.22	11.36 ± 4.85							
							PT	F/U				
Smitherman [32] (2016)	HA Frequency (days/month)	I	22.7	16.6	11.6		*p* = 0.883	*p* = 0.028			N	○ Headache frequency reduction from baseline of I group was not statistically significant com-pared with C group at PT, FU *p* = 0.883 ○ I and C group showed clinically meaningful re-ductions in MIDAS, HIT-6, headache severity at PT, FU with no group differences when controlling for baseline scores ○ No significant group difference in Headache frequency, HIT-6, MIDAS score, headache severity, ESS, PHQ-9, GAD-7, CEQ ○ Significant group difference in TST, sleep efficiency, PSQI
		C	19.6	12.5	14.7							
	MIDAS	I	59.9 ± 39.0	44.2 ± 43.1	31.9 ± 33.2		N					
		C	54.5 ± 41.0	41.0 ± 46.2	34.7 ± 34.5							
	HIT-6	I	66.9 ± 3.8	62.6 ± 5.3	59.9 ± 5.5		N					
		C	64.8 ± 3.9	61.4 ± 8.0	59.6 ± 7.2							
	HA Severity	I	5.2 ± 0.9	5.1 ± 1.4	4.5 ± 1.5		N					
		C	5.4 ± 1.6	5.2 ± 2.1	5.1 ± 1.9							
	PSQI ^7^	I	11.3 ± 4.4	7.6 ± 2.6	7.0 ± 3.1		*p* = 0.009					
		C	11.6 ± 2.5	10.9 ± 3.8	11.5 ± 3.9							
	ESS ^8^	I	11.0 ± 3.4	9.0 ± 3.2	8.9 ± 3.55		N					
		C	9.9 ± 4.892	9.2 ± 4.7	8.8 ± 4.6							
	TST (h) ^9^	I	7.4 ± 1.5	7.3 ± 1.4	8.3 ± 2.6		*p* = 0.049					
		C	6.7 ± 1.5	6.9 ± 1.2	6.8 ± 0.5							
	Sleep Efficiency	I	81.2 ± 7.7	79.1 ± 8.9	84.9 ± 4.5		*p* = 0.001					
		C	81.2 ± 8.3	82.4 ± 6.4	80.9 ± 4.9							
	PHQ-9 ^10^	I	12.1 ± 5.8	6.9 ± 4.8	6.3 ± 4.6		*p* = 0.054					
		C	10.5 ± 4.5	8.4 ± 4.7	8.6 ± 4.7							
	GAD-7 ^11^	I	10.6 ± 6.4	6.6 ± 5.2	6.3 ± 4.8		*p* = 0.430					
		C	9.8 ± 5.3	7.0 ± 4.6	6.9 ± 4.9							
Cousins [33] (2015)					4 M						N	○ At 4 months after treatment, no significant change between I and C group statistically in. - Diary headache days - Medication days/ month - MIDAS, HIT-6, HADS-A, - HADS-D, IPQ
	HA days/month	I	12.03 ± 8.70		9 ± 7.27		N					
		C	11.54 ± 6.64		9.68 ± 6.28							
	Using rescue medication days/month	I	6.69 ± 5.30		5.86 ± 5.12		N					
		C	7.08 ± 5.87		6.2 ± 4.86							
	MIDAS	I	51.03 ± 43.68		33.86 ± 34.93		N					
		C	65.78 ± 46.79		53.85 ± 78.49							
	HIT-6 ^12^	I	66.5 ± 5.88		59.17 ± 8.19		N					
		C	65.97 ± 4.41		60.85 ± 8.4							
	HADS-A ^13^	I	7.78 ± 4.01		5.76 ± 4.45		N					
		C	9.32 ± 3.55		7.96 ± 4.37							
	HADS-D	I	5.83 ± 4.61		4.24 ± 4.6		N					
		C	5.68 ± 3.09		4.52 ± 3.51							
	Brief IPQ ^14^	I	52.81 ± 9.69		44.17 ± 15.89		N					
		C	51.41 ± 9.77		45.26 ± 10.17							
Wells [12] (2014)							PT	FU			N	○ The severity and du-ration of all head-aches decreased in the I group, but not statistically significant ○ Significant decrease in I group compared with C group on HIT-6 at PT (*p* = 0.043), FU(*p* = 0.022) and MIDAS at PT (*p* = 0.017) ○ Self-efficacy and mindfulness also in-creased at PT (*p* = 0.035) ○ MBSR is safe and feasible for adults with migraine.
	Migraine Frequency /month	I	4.2 ± 2.9	2.6 ± 3.8			*p* = 0.38	*p* = 0.63				
		C	2.9 ± 5.2	2.7 ± 6.9								
	HA Frequency days/month	I	9.9 *	9.0 *	9.0 *		*p* = 0.14	*p* = 0.22				
		C	12.3 *	10.0 *	7.7 *							
	HA severity (0–10)	I	4.4 *	3.2 *	3.3 *		*p* = 0.053	*p* = 0.66				
		C	4.8 *	5.2 *	4.8 *							
	HA duration	I	5.1 *	2.9 *	3.6 *		*p* = 0.043	*p* = 0.19				
		C	6.4 *	6.1 *	6.1 *							
	HIT-6	I	63.0 ± 8.0	57.6 ± 6.7	58.3 ± 6.0		*p* = 0.043	*p* = 0.022				
		C	64.7 ± 5.0	64.1 ± 3.8	64.1 ± 3.9							
	MIDAS	I	12.5 ± 9.8	5.9 ± 5.3	5.8 ± 3.8		*p* = 0.017	*p* = 0.072				
		C	11.0 ± 6.7	17.0 ± 11.9	12.0 ± 8.1							
	HA Management Self Efficacy	I	117.2 ± 18.7	122.6 ± 25.0	124.5 ± 22.6		*p* = 0.035	*p* = 0.060				
		C	118.4 ± 31.1	110.7 ± 29.2	111.9 ± 35.7							
	Five Factor Mindfulness	I	142.9 ± 14.7	149.1 ± 18.7	153.8 ± 19.7		*p* = 0.035	*p* = 0.045				
		C	143.7 ± 20.3	136.8 ± 18.3	138.0 ± 19.6							
	MSQoL ^15^	I	47.0 *	31.5 *	38.1 *		*p* = 0.12	*p* = 0.035				
		C	46.4 *	45.2 *	45.2 *							
	PHQ-9 ^16^	I	3.6 ± 3.0	2.0 ± 1.8	2.7 ± 2.2		*p* = 0.77	*p* = 0.59				
		C	6.4 ± 6.5	4.2 ± 1.8	3.9 ± 1.9							
	STAI ^17^	I	68.7 ± 16.3	61.6 ± 15.0	60.5 ± 16.8		*p* = 0.13	*p* = 0.10				
		C	67.0 ± 15.8	70.2 ± 14.9	69.1 ± 10.3							
	Perceived Stress Scale-10	I	15.8 ± 6.4	13.3 ± 5.1	12.1 ± 5.1		*p* = 0.87	*p* = 0.27				
		C	14 ± 8.1	12.1 ± 8.0	13.6 ± 7.0							
Rapoff [34] (2014)					3 M		BL	PT			N	○ In I group, pain severity decreased at PT compared with C group (*p* = 0.03) ○ At 3 M FU, significant change in PedMIDAS score in I group compared with C group (*p* = 0.04) No other group differences at PT or 3M FU
	HA frequency (% of days)	I	41.09 ± 22.67	31.28 ± 28.14	21.43 ± 23.47		*p* = 0.48	*p* = 0.46		*p* = 0.36		
		C	40.67 ± 28.79	32.14 ± 22.23	18.18 ± 17.60							
	HA duration (hr/episode)	I	5.47 ± 4.20	4.47 ± 4.26	1.53 ± 0.91		*p* = 0.19	*p* = 0.24		*p* = 0.07		
		C	4.15 ± 3.88	5.56 ± 4.01	4.25 ± 5.19							
	HA severity (VAS)	I	5.06 ± 1.84	5.06 ± 1.50	4.46 ± 1.88		*p* = 0.07	*p* = 0.03		*p* = 0.20		
		C	6.00 ± 1.52	6.25 ± 1.92	3.68 ± 2.04							
	PedMIDAS total ^18^	I	13.26 ± 9.69	7.82 ± 10.59	0.91 ± 1.45		*p* = 0.25	*p* = 0.14		*p* = 0.05		
		C	15.53 ± 10.08	12.29 ± 12.94	3.50 ± 4.86							
	PedsQL total ^19^	I	82.10 ± 12.18	83.70 ± 12.07	84.88 ± 18.22		*p* = 0.25	*p* = 0.26		*p* = 0.46		
		C	79.35 ± 11.55	80.69 ± 14.36	85.67 ± 14.32							
Fritsche [35] (2010)					3 M	12–30 M					N	○ Significant change in time effect observed in I, C group in headache days, migraine days, Intake at headache days, Intake at migraine days (*p* < 0.001) ○ Improvement in psychological variables (*p* < 0.001) - Remained stable in both groups at short- and long-term F/U ○ MCT (C) and biblio-therapy (I) are useful to prevent “medication overuse headache” and transition to chronic head-ache
	HA days	I	11.40 ± 5.92	9.17 ± 5.45	8.55 ± 5.51	8.68 ± 5.29	Time effect *p* < 0.001					
		C	10.51 ± 4.98	8.47 ± 5.54	8.11 ± 4.82	8.33 ± 5.15						
	Migraine days	I	7.23 ± 3.70	5.60 ± 3.79	6.15 ± 3.97	6.15 ± 4.02	Time effect *p* < 0.001					
		C	7.27 ± 3.82	5.78 ± 4.01	5.45 ± 3.16	5.84 ± 3.76						
	HA disability	I	4.46 ± 1.80	4.49 ± 2.01	4.61 ± 1.97	4.39 ± 2.16	Time effect N.S					
		C	4.16 ± 1.56	4.13 ± 1.97	4.25 ± 1.88	4.40 ± 1.73						
	Intake at HA days	I	7.17 ± 2.48	5.92 ± 3.10	5.93 ± 3.23	6.18 ± 3.65	Time effect *p* < 0.001					
		C	7.58 ± 3.11	6.35 ± 3.66	6.47 ± 3.20	6.00 ± 2.82						
	Intake at migraine days	I	5.27 ± 2.25	4.30 ± 2.76	4.83 ± 3.00	5.03 ± 3.52	Time effect *p* < 0.001					
		C	6.25 ± 2.98	5.04 ± 3.11	4.75 ± 2.82	5.02 ± 2.78						
Calhoun [36] (2007)	HA frequency /28 days	I	24.2 **	17.4 **			*p* = 0.001				N	○ In I group, statistically significant reduction compared to C group observed headache frequency (*p* = 0.001) and Headache intensity(*p*= 0.01) at PT ○ No one in C group re-verted to episodic migraine, and 48.5% in I group reverted to epi-sodic migraine
		C	23.2 **	23.9 **								
	HA Intensity	I	46.7 **	28.3 **			*p* = 0.01					
		C	50.2 **	44.1 **								
	Reverted to episodic migraine						*p* = 0.029					

Scharff [37] (2002)							PT		F/U		N	○ Clinical improvement in all variables (head-ache index change, highest intensity, days with headache) over time and compared with C group. ○ In HWB and HCB group ○ At PT, there is no significant temperature change ○ At 3 M F/U, temperature changes in both HWB and HCB group were significant com-pared to BL ○ At 6 M F/U, HWB group showed clinical improvement compared to HCB group
	HA Index change	(1) Effect of time (Pillai’s trace = 0.267, F[3, 29] = 3.53, *p* < 0.03) (2) Trend for treatment group (Pillai’s trace = 0.36, F[6, 60] = 2.21, *p* < 0.05)		(1) Effect for time (Pillai’s trace =0.81, F[9, 12] = 5.62, *p* < 0.01) (2) Trend for treatment group (Pillai’s trace = 0.32, F[3, 18] = 2.80, *p* < 0.07)			*p* < 0.005		*p* < 0.001			
	Highest intensity rating for 2-week						*p* < 0.01		N			
	HA days						*p* < 0.02		*p* < 0.01			
	Temperature change	(1) Effect of time (Philai’s trace = 0.44, F[12, 69] = 4.44, *p* < 0.001) (2) No significant difference in treatment			6 M		N					
				72.2% of HWB ^20^, 33.3% of HCB ^21^ were significant compared to BL (χ² [1] = 3.76, *p* < 0.05).	100% of HWB 62.5% of HCB showed clinical improvement (χ² [1] = 4.50, *p* < 0.05).							

^1^ HDI: Headache Disability Inventory; ^2^ MIDAS: Migraine Disability Asessment; ^3^ MIDAS A: Self-reported headache days over a 90 day period, divided by 3; ^4^ MIDAS B: Self-reported average headache attack intensity over a 90 day period (1–10); ^5^ Average Attack Intensity/30 Days (by 1–3 scale); ^6^ MIDI = Migraine Disability Index (by 0–10 scale); ^7^ PSQI: Pittsburgh Sleep Quality Index; ^8^ ESS: Epworth Sleepiness Scale; ^9^ TST: Total Sleep Time; ^10^ PHQ-9: Patient Health Questionnaire 9-item Depression Module; ^11^ GAD-7: Generalized Anxiety Disorder 7-item Scale; ^12^ HIT-6: Headache Impact Test; ^13^ HADS: Hospital Anxiety and Depression Scale; ^14^ Brief-IPQ: Brief Illness Perceptions Questionnaire; ^15^ MSQoL: Migraine-Specific Quality of Life; ^16^ PHQ-9: Patient Health Questionnaire-depression module; ^17^ STAI: State Trait Anxiety Inventory; ^18^ PedMIDAS: The Pediatric Migraine Disability Assessment; ^19^ PedsQL: Pediatric Quality of Life Inventory (4th ed.); ^20^ HWB: handwarming biofeedback; ^21^ HCW: handcooling biofeedback; I, intervention; C, control; N.S., not statistically significant; BL, baseline; PT, post-treatment F/U, follow-up; M, months; Yrs, years; HA, headache; CBT, cognitive behavioral therapy; MCT, minimal contact therapy; Frequency of migraine, days per month during treatment; T, time effect.

## Data Availability

The datasets generated during and/or analyzed during the current study are available from the corresponding author on reasonable request.

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
