# Peer review of "Cognitive Behavioral Therapy for Migraine Headache: A Systematic Review and Meta-Analysis"

_medicina, 2021, doi:10.3390/medicina58010044_

Round 1
Reviewer 1 Report
My questions were answered and I have no further comment
Author Response
We revised our manuscript based on your advise.

Reviewer 2 Report
As it was previously recommended, a systematic review should be undertaken in case clinical and methodological heterogeneity impede data synthesis.
Author Response
We rewrite our manuscirpt based on your advise.
Round 2
Reviewer 2 Report
Statistical analysis is both redundant and wrong. As I suggested earlier, a systematic review should be undertaken (it is not appropriate to pool ‘apples and oranges’ together, e.g., treatment as usual is totally different than placebo, so when you pool these groups together you end up with an incomprehensible result, that is experimental intervention is more efficient than what????? Placebo? Usual treatment? The correct answer is neither). FE model is repeatedly used in pooling highly heterogeneous results (even when I2 > 75%). This violates even your lenient cut-off of 75% (too lenient, please use the 30% threshold, as well as the Q statistic). Moreover, please clinically address heterogeneity. The statistical approach (implementation of RE model) is accepted as a last solution (when studies appear to be identical in terms of methodology and patient characteristics, and among study differences cannot be clinically explained). Authors can usually detect potential reasons for statistical heterogeneity. In case you can detect such reasons, you should report them and avoid pooling clinically heterogeneous studies. In case clinical heterogeneity is not detected you may pool these results using RE model (as stated above there is substantial clinical heterogeneity that you should not neglect).
Figures: Forest-plots require extensive editing. X-axis scales are presented at MD values between -100 and 100, while MD range between -10 and 10 (this leads to a loss of the visual impact a forest plot has on the reader). 'Experimental' and 'control' groups should be replaced by the exact interventions.
Again, RoB was insufficiently explored. Lets takefor example the study of Powers et al. You mention that ''In I group, headache days and PedMIDAS decreased significantly. â—‹ Headache days were ≥50% Reduction in 66% of I and 36% of C at PT, 86% of I and 69% of C at F/U. â—‹ PedMIDAS <20 points were in 75% of I, and 56% of C at PT, 88% of I and 76% of C at F/U.'' A RCT is performed so that we compare one experimental intervention, with one already tested or placebo intervention. If our aim was to examine an intervention on its own, then randomization would not be necessary. So, why perform an RCT if we are to report our findings only for the experimental group? The answer is that authors selectively report these findins in order to achieve publication (insignificant findings are more difficult to publish). So, how come you did not detect publication bias in this (many other as well) study?
Author Response
(Reviewer’s comment 1)Statistical analysis is both redundant and wrong. As I suggested earlier, a systematic review should be undertaken (it is not appropriate to pool ‘apples and oranges’ together, e.g., treatment as usual is totally different than placebo, so when you pool these groups together you end up with an incomprehensible result, that is experimental intervention is more efficient than what????? Placebo? Usual treatment? The correct answer is neither). FE model is repeatedly used in pooling highly heterogeneous results (even when I2 > 75%). This violates even your lenient cut-off of 75% (too lenient, please use the 30% threshold, as well as the Q statistic). Moreover, please clinically address heterogeneity. The statistical approach (implementation of RE model) is accepted as a last solution (when studies appear to be identical in terms of methodology and patient characteristics, and among study differences cannot be clinically explained). Authors can usually detect potential reasons for statistical heterogeneity. In case you can detect such reasons, you should report them and avoid pooling clinically heterogeneous studies. In case clinical heterogeneity is not detected you may pool these results using RE model (as stated above there is substantial clinical heterogeneity that you should not neglect).
Answer) We have reclassified the studies used in subgroup analysis according to your advice. As a control group, it was largely divided into studies that included only educational treatment, studies with weight list, treatment asual, standard medical care, studies using vibliotherapy, and sham treatment, and finally 3arm studies using both CBT and weight list.
As for heterogeneity, we are also aware that setting a 30% limit as you suggested helps reduce and analyze heterogeneity among studies. Therefore, we reclassified the group according to your advice to reduce clinical heterogeneity as much as possible. However, this study was conducted in connection with the clinical practice guidline, and it would be difficult to change the I2 value by setting it at 75 at the beginning of the study. After dividing the group again, there was no study showing high heterogeneity other than the subgroup analysis of headache frequency compared to the WL/TAU/SMC group, so the subgroup analysis was changed to random effect model according to your advice.
(Reviewer’s comment 2) Figures: Forest-plots require extensive editing. X-axis scales are presented at MD values between -100 and 100, while MD range between -10 and 10 (this leads to a loss of the visual impact a forest plot has on the reader). 'Experimental' and 'control' groups should be replaced by the exact interventions.
Answer) We revised the figures according to your advice.
(Reviewer’s comment 3) Again, RoB was insufficiently explored. Lets takefor example the study of Powers et al. You mention that ''In I group, headache days and PedMIDAS decreased significantly. â—‹ Headache days were ≥50% Reduction in 66% of I and 36% of C at PT, 86% of I and 69% of C at F/U. â—‹ PedMIDAS <20 points were in 75% of I, and 56% of C at PT, 88% of I and 76% of C at F/U.'' A RCT is performed so that we compare one experimental intervention, with one already tested or placebo intervention. If our aim was to examine an intervention on its own, then randomization would not be necessary. So, why perform an RCT if we are to report our findings only for the experimental group? The answer is that authors selectively report these findins in order to achieve publication (insignificant findings are more difficult to publish). So, how come you did not detect publication bias in this (many other as well) study?
Answer) We rechecked the RoB and found one detectable risk. In the article of Grazzi, we found that MIDAS and HIT-6, HADS, PCS were assessed but not reported. By the way, there were too many omissions to understand the results of the paper, so we added the missing contents based on your advice.
This manuscript is a resubmission of an earlier submission. The following is a list of the peer review reports and author responses from that submission.
Round 1
Reviewer 1 Report
The paper Cognitive behavioral therapy for migraine headache 2 systematic review and meta-analysis by Ji-yong Bae, KMD et al. conducted a meta-analysis of CBT treatment for migraine. The subject is impotent since migraine patient's often cannot find cure in medications. However the paper has few imitations.
Absract
It is mentioned few timed about the autonomic nervous system involvement – What do the authors mean?
Introduction
Line 50 – see above
Line 51 – The sentence has no meaning
The introduction should include; description of abortive versus preventive treatment. Line 60 - The first time CBT is mentioned it should not be as an acronym.
Methods
What was the patients' age?
Was the migraine chronic or episodic?
Results
Figure 1 – In the screening – n+233 should not be mentioned twice.
Discussion
The discussion is written poorly. It should begin with the study findings, than it should include details of the various preventive medication treatments available and their shortcuts along with other alternative treatments. The authors should compare the outcome of CBT to other various treatments.
Bibliography
Papers with many authors (like #2) should be shorten to et al.

Author Response
Authors’ Response to the Review Comments
We appreciate the time given and efforts made by the editor and referees in reviewing this paper. We have addressed all issues indicated in the review report in a point by point manner, and changed those parts in RED. We hope that the revised paper will be able to meet the journal publication requirements.
Once again, we very much appreciate you taking the time to comment on our manuscript.
Response to Comments
Reviewer 1
(Reviewer’s comment 1) Abstract It is mentioned few timed about the autonomic nervous system involvement – What do the authors mean?
Answer ) We mean migraine headaches can reduce the quality of life due to frequent symptoms of the autonomic nervous system as well as moderate headaches. I revised the contents to clarify what I wanted to talk about according to the advice.
Answer) We clarified the contents of abortive and preventive treatment for migraines and revised them by adding insufficient contents.
(Reviewer’s comment 4) Line 60 - The first time CBT is mentioned it should not be as an acronym.
(Reviewer’s comment 5) What was the patients' age? Was the migraine chronic or episodic?
(Reviewer’s comment 6) Figure 1 – In the screening – n+233 should not be mentioned twice.
Answer) According to the reviewer's advice, we removed the duplicated part and corrected it.
(Reviewer’s comment 7) Papers with many authors (like #2) should be shorten to et al.
Answer ) We revised the contents of the reference as recommended.

Reviewer 2 Report
The present article discusses the use of cognitive behavioral therapy (CBT) for the prevention of migraine, an overall very interesting subject. Extensive English editing is required (throughout the text) while the following major issues must be attended to.
HA: Headache: please avoid abbreviating single words
Abstract: Result
‘‘Among 373 studies, 4 randomized controlled trials were finally included in this systematic review. 9 studies. 7 out of 11 Randomized controlled trials (RCTs) were conducted in USA and 3 were held in UK, Germany, Iran and Italy, respectively.’’ This section is completely incomprehensible. 4 articles, 9 articles, 11 articles, 7 plus 3 articles… Please be consistent.
‘‘In subgroup analysis, the headache strength was statistically significantly reduced.’’ Please either define subgroup analysis or avoid mentioning its results in the abstract.
Introduction
‘‘Preferred treatment for migraine medication.’’ A sentence without a verb is not a standalone sentence. Please be more careful with your writing.
‘‘Although CBT has been used as one of the treatments of migraine headaches, there was only research on pediatric migraine and no research has been conducted on all ages.’’ Then what is your article discussing? I found several articles for CBT in migraine prevention in adults.
Please cite the following landmark articles when discussing the use of complementary treatments:
- Endogenous Melatonin Levels and Therapeutic Use of Exogenous Melatonin in Migraine: Systematic Review and Meta-Analysis. Headache. 2020
- Prophylaxis of migraine headaches with riboflavin: A systematic review. J Clin Pharm Ther. 2017
- Pyridoxine, folate and cobalamin for migraine: A systematic review. Acta Neurol Scand. 2020
- Zeng Z, Li Y, Lu S, Huang W, Di W. Efficacy of CoQ10 as supplementation for migraine: A meta-analysis. Acta Neurol Scand. 2019
- Vitamin D serum levels in patients with migraine: A meta-analysis. Rev Neurol (Paris). 2020
Eligibility criteria
‘‘Eligible participants were migraine patients who were satisfying migraine criteria. Most of migraine Eligibility criteria International Society (IHS). If there were other criteria, we discussed criteria with ICHD and HIS and similar conditions were included.’’ First, this section should be edited to make better sense. Second, eligibility criteria should be more detailed. For instance, describe in more detail the population of interest: adult or non-adult, with chronic and/or frequent migraine, with and/or without aura, with and/or without MOH, etc. Also, describe the eligibility of study designs (RCTs? Parallel? Active and sham comparators?) Any other details of interest should be included (duration of interventions if applicable, time between intervention and outcome assessment, etc).
Assessment of risk of bias
‘‘Deflection Risk (ROB) of Cochrane was used for quality assessment of the included study.’’ There is not a deflection risk of Cochrane, but there is a risk of bias assessment tool, if this is what you mean. Also, your description of the tool is deficient. Please cite: Intravenous sodium valproate in status epilepticus: review and Meta-analysis. Int J Neurosci. 2021, for a detailed description of the RoB tool.
Data synthesis and meta-analysis
Please pay attention to the following matters: First, clinical heterogeneity is more important than statistical heterogeneity. Herein, the potential combination of studies is clinically filtered first. We cannot, for example, pool results from studies with different comparators or different CBT methods (because our conclusions will be incomprehensible, i.e., CBT will be inferior or equally effective or superior to what???, similarly which CBT method is inferior or equally effective or superior to the comparator???). Differences in CBT methodology and active or sham comparators are usually reflected into high heterogeneity. Therefore, when finding high statistical heterogeneity, one should first try to explain this and then decide if it is appliable to use RE models for pooling. In this context, please refer to the statistical plan of the article: Serum lipid abnormalities in migraine: A meta-analysis of observational studies. Headache. 2021, and apply the 30% I2 cut-off as described by the authors. Stricter statistical criteria usually increase the probability to detect heterogeneity among studies.
‘‘Since we expected that different outcome assessment tools would be used in most studies, we assessed effect estimates with a risk ratio (RR) for dichotomous outcomes and the standardized mean difference (SMD) for continuous outcomes.’’ Outcome measures should be defined a-priori. Assumptions for the potential assessed outcomes among the retrieved studies are prone to multiple and selective testing that may confuse trivial associations as significant. Please clearly define outcome measures and their measurement methods (scale or categorical) in this section. It is important to abide by the IHS guidelines when defining outcome measures. In particular, please avoid to use within group differences as potential indications of efficacy and employ between group differences (post – pre intervention per group).
Also, this section lacks detail. Age of the participants, type of migraine (with or without aura, frequent or chronic, etc), time of follow-up investigations (one week or one month or one year post interventional assessments should not be pooled together), similarities of comparators, etc, assume an important role in the homogeneity of studies. They should be taken into consideration when pooling findings from different articles. For instance, adult studies should be separately presented from non-adult articles. Please provide a much clearer and well-defined statistical plan, according to my suggestions. In case data synthesis is inappropriate, please base your conclusions on the systematic review of published articles (avoid the meta-analysis).
The Results of Literature Search and Screening
‘‘55 articles without full-text were removed’’. This is an unacceptable practice. Important selection bias will be introduced. Please contact corresponding authors or resort to an experienced librarian so that you can obtain these full texts.
Risk of bias in Included Studies
While scanning your tables I found selective reporting among retrieved articles. How come you have not detected this bias. When referring to the RoB tool, each item should be assessed for every defined outcome as defined by the authors of the meta-analysis and not by the authors of each retrieved article (multiple RoB figures are usually required, however you can provide one RoB figure for the primary outcome alone or - conventionally but not quite right - one RoB figure for the worst risk assessments per item regarding all defined outcomes). When retrieved findings are not applicable to translation according to your definitions, selective reporting bias exists regarding defined outcomes. I hope this makes it clear to you.
Outcomes
‘‘6 RCTs[21,25-27,29,31] had reported the headache frequency, excluding 5 studies (3 studies[28,32-33] that did not mention mean, standard deviation, and 2 studies[12,30] that did not mention).’’Your description is incomprehensible. ‘‘2 studies that did not mention’’ what??? Please be specific and clear.
‘‘The days of headache per month decreased significantly compared with control group, and unfortunately high heterogeneity (?2 = 28.18, P = 0.0003, I 2 = 82%)’’ Again, this is not an understandable sentence (and unfortunately high heterogeneity????).
In this part of the text pay attention to the pooling advice I provide above. It is important to avoid pooling results from different comparisons.
Discussion
According to my suggestion please modify this part based on your new findings.
Author Response
Authors’ Response to the Review Comments
We appreciate the time given and efforts made by the editor and referees in reviewing this paper. We have addressed all issues indicated in the review report in a point by point manner, and changed those parts in RED. We hope that the revised paper will be able to meet the journal publication requirements.
Once again, we very much appreciate you taking the time to comment on our manuscript.
Reviewer2
(Reviewer’s comment 1) HA: Headache: please avoid abbreviating single words
Answer) According to your comment, we revised HA to headache
(Reviewer’s comment 2) Abstract: Result
‘‘Among 373 studies, 4 randomized controlled trials were finally included in this systematic review. 9 studies. 7 out of 11 Randomized controlled trials (RCTs) were conducted in USA and 3 were held in UK, Germany, Iran and Italy, respectively.’’ This section is completely incomprehensible. 4 articles, 9 articles, 11 articles, 7 plus 3 articles… Please be consistent.
Answer ) “7 RCTs[12,21,25,28,30,32-33] were conducted in USA, 1 RCT[29] were conducted in UK, 1 RCT[26] were conducted in Italy, 1 RCT[31] were conducted in Germany and 1 RCT[27] were conducted in Iran.” The above is classified according to the country in which the study was conducted. We deleted the confusing contents of Figure 1.
(Reviewer’s comment 3) ‘‘In subgroup analysis, the headache strength was statistically significantly reduced.’’ Please either define subgroup analysis or avoid mentioning its results in the abstract.
Answer ) We deleted the contents about subgroup analysis
Introduction
(Reviewer’s comment 4) ‘‘Preferred treatment for migraine medication.’’ A sentence without a verb is not a standalone sentence. Please be more careful with your writing.
Answer ) We deleted the sentence
(Reviewer’s comment 5) ‘‘Although CBT has been used as one of the treatments of migraine headaches, there was only research on pediatric migraine and no research has been conducted on all ages.’’ Then what is your article discussing? I found several articles for CBT in migraine prevention in adults.
Answer ) Most of the studies were randomized controlled trials of small samples or case reports I revised it because I thought replacing the word with "systematic review" could clarify the meaning more than "Research" after listening to your advice.
(Reviewer’s comment 6) Please cite the following landmark articles when discussing the use of complementary treatments:
- Endogenous Melatonin Levels and Therapeutic Use of Exogenous Melatonin in Migraine: Systematic Review and Meta-Analysis. Headache. 2020
- Prophylaxis of migraine headaches with riboflavin: A systematic review. J Clin Pharm Ther. 2017
- Pyridoxine, folate and cobalamin for migraine: A systematic review. Acta Neurol Scand. 2020
- Zeng Z, Li Y, Lu S, Huang W, Di W. Efficacy of CoQ10 as supplementation for migraine: A meta-analysis. Acta Neurol Scand. 2019
- Vitamin D serum levels in patients with migraine: A meta-analysis. Rev Neurol (Paris). 2020
Answer ) We added the studies you recommended to explain the supplementary therapy for migraine treatment.
Eligibility criteria
(Reviewer’s comment 7) ‘‘Eligible participants were migraine patients who were satisfying migraine criteria. Most of migraine Eligibility criteria International Society (IHS). If there were other criteria, we discussed criteria with ICHD and HIS and similar conditions were included.’’ First, this section should be edited to make better sense. Second, eligibility criteria should be more detailed. For instance, describe in more detail the population of interest: adult or non-adult, with chronic and/or frequent migraine, with and/or without aura, with and/or without MOH, etc. Also, describe the eligibility of study designs (RCTs? Parallel? Active and sham comparators?) Any other details of interest should be included (duration of interventions if applicable, time between intervention and outcome assessment, etc).
Answer ) Based on your advice, we added information on the type of studies included, type of migraine, age, gender, and outcomes.
Assessment of risk of bias
(Reviewer’s comment 8) ‘‘Deflection Risk (ROB) of Cochrane was used for quality assessment of the included study.’’ There is not a deflection risk of Cochrane, but there is a risk of bias assessment tool, if this is what you mean. Also, your description of the tool is deficient. Please cite: Intravenous sodium valproate in status epilepticus: review and Meta-analysis. Int J Neurosci. 2021, for a detailed description of the RoB tool.
Data synthesis and meta-analysis
(Reviewer’s comment 9) Please pay attention to the following matters: First, clinical heterogeneity is more important than statistical heterogeneity. Herein, the potential combination of studies is clinically filtered first. We cannot, for example, pool results from studies with different comparators or different CBT methods (because our conclusions will be incomprehensible, i.e., CBT will be inferior or equally effective or superior to what???, similarly which CBT method is inferior or equally effective or superior to the comparator???). Differences in CBT methodology and active or sham comparators are usually reflected into high heterogeneity. Therefore, when finding high statistical heterogeneity, one should first try to explain this and then decide if it is appliable to use RE models for pooling. In this context, please refer to the statistical plan of the article: Serum lipid abnormalities in migraine: A meta-analysis of observational studies. Headache. 2021, and apply the 30% I2 cut-off as described by the authors. Stricter statistical criteria usually increase the probability to detect heterogeneity among studies.
Answer ) As you said, we understand that clinical heterogeneity is important because participants and intervention may differ from study to study when evaluating heterogeneity. However, in the case of CBT, there was no standardized method, and there was a difference in CBT method among researchers. It would be nice to compare studies with less clinical heterogeneity, but CBT did not have obvious sham control, so it was not possible to contrast like the comparison between acupuncture and sham acupuncture. In future studies, we will accept your advice and consider clinical heterogeneity. The lack of consideration for clinical heterogeneity will be described in more detail in the limitations of this study. Thank you.
(Reviewer’s comment 10) ‘‘Since we expected that different outcome assessment tools would be used in most studies, we assessed effect estimates with a risk ratio (RR) for dichotomous outcomes and the standardized mean difference (SMD) for continuous outcomes.’’ Outcome measures should be defined a-priori. Assumptions for the potential assessed outcomes among the retrieved studies are prone to multiple and selective testing that may confuse trivial associations as significant. Please clearly define outcome measures and their measurement methods (scale or categorical) in this section. It is important to abide by the IHS guidelines when defining outcome measures. In particular, please avoid to use within group differences as potential indications of efficacy and employ between group differences (post – pre intervention per group).
Answer ) The content was incorrectly written in the process of translating the sentence. I revised the contents based on your advice.
(Reviewer’s comment 11) Also, this section lacks detail. Age of the participants, type of migraine (with or without aura, frequent or chronic, etc), time of follow-up investigations (one week or one month or one year post interventional assessments should not be pooled together), similarities of comparators, etc, assume an important role in the homogeneity of studies. They should be taken into consideration when pooling findings from different articles. For instance, adult studies should be separately presented from non-adult articles. Please provide a much clearer and well-defined statistical plan, according to my suggestions. In case data synthesis is inappropriate, please base your conclusions on the systematic review of published articles (avoid the meta-analysis
Answer ) As I answered the previous advice, there was a little difficulty in designing the characteristics of the intervention(CBT), so it seems that consideration for clinical heterogeneity is insufficient. We will conduct better research by referring to future research.
(Reviewer’s comment 12) The Results of Literature Search and Screening
‘‘55 articles without full-text were removed’’. This is an unacceptable practice. Important selection bias will be introduced. Please contact corresponding authors or resort to an experienced librarian so that you can obtain these full texts.
Answer ) 55 studies with no full text refer to papers that have not been obtained through the library. This paper was mainly searched on CNKI, and the text was marked as unavailable.
Risk of bias in Included Studies
(Reviewer’s comment 13) While scanning your tables I found selective reporting among retrieved articles. How come you have not detected this bias. When referring to the RoB tool, each item should be assessed for every defined outcome as defined by the authors of the meta-analysis and not by the authors of each retrieved article (multiple RoB figures are usually required, however you can provide one RoB figure for the primary outcome alone or - conventionally but not quite right - one RoB figure for the worst risk assessments per item regarding all defined outcomes). When retrieved findings are not applicable to translation according to your definitions, selective reporting bias exists regarding defined outcomes. I hope this makes it clear to you.
Answer ) Of the 11 final studies included, we discussed that the paper of Licia Grazzi was a high risk corresponding to duplication publication bias, but we couldn't find any previous research data and assigned it to unclear risk. We listened to your advice and we thought about the bias again, and adjusted the ROB evaluation for the study to high risk. If you give me an additional view on risk of bias, I will discuss and revise it.
Outcomes
(Reviewer’s comment 14) ‘‘6 RCTs[21,25-27,29,31] had reported the headache frequency, excluding 5 studies (3 studies[28,32-33] that did not mention mean, standard deviation, and 2 studies[12,30] that did not mention).’’Your description is incomprehensible. ‘‘2 studies that did not mention’’ what??? Please be specific and clear.
Answer ) After listening to your advice, we added an object to clarify why 2 studies were not included in the analysis.
(Reviewer’s comment 15) ‘‘The days of headache per month decreased significantly compared with control group, and unfortunately high heterogeneity (?2 = 28.18, P = 0.0003, I 2 = 82%)’’ Again, this is not an understandable sentence (and unfortunately high heterogeneity????).
Answer ) We revised the sentence to have the appropriate meaning we want to convey.
In this part of the text pay attention to the pooling advice I provide above. It is important to avoid pooling results from different comparisons.
Discussion
(Reviewer’s comment 16) According to my suggestion please modify this part based on your new findings.
Answer ) we revised our discussion after listening your advice
